# The Role of Oral Contraceptive Pills in Hidradenitis Suppurativa: A Cohort Study

**DOI:** 10.3390/life11070697

**Published:** 2021-07-15

**Authors:** Trinidad Montero-Vilchez, Andrea Valenzuela-Amigo, Carlos Cuenca-Barrales, Salvador Arias-Santiago, Ana Leyva-García, Alejandro Molina-Leyva

**Affiliations:** 1Dermatology, Hidradenitis Suppurativa Clinic, Hospital Universitario Virgen de las Nieves, 18012 Granada, Spain; tmonterov@correo.ugr.es (T.M.-V.); carloscuenca1991@gmail.com (C.C.-B.); alejandromolinaleyva@gmail.com (A.M.-L.); 2Instituto de Investigación Biosanitaria Granada, 18012 Granada, Spain; 3Nursing Department, Faculty of Nursing, University of Granada, 18012 Granada, Spain; tricarmv@hotmail.com (A.V.-A.); amleyva@ugr.es (A.L.-G.); 4Dermatology Department, Faculty of Medicine, University of Granada, 18016 Granada, Spain; 5European Hidradenitis Suppurativa Foundation (EHSF), 06847 Dessau-Roßlau, Germany

**Keywords:** contraceptives, hidradenitis suppurativa, premenstrual syndrome

## Abstract

There is a need to establish the role of antiandrogens as an alternative or concomitant therapy for hidradenitis suppurativa (HS). Thus, the objectives of this study are (1) to assess the effectiveness of oral contraceptive pills (OCPs) at week 12 in HS women, and (2) to describe the clinical profile of patients receiving oral contraceptive pills (OCPs). A prospective observational study was designed. This study included 100 participants, 50 women with HS who started OCPs for the first time at our HS Clinic and 50 participants without OCP treatment. The main outcome of interest was the percentage of reduction in total abscess and inflammatory nodule (AN) count at week 12. Thirty-three women received combined OCPs and 17 non-combined OCP. HS patients with OCPs treatment were younger (31.7 vs. 40.9 years, *p* < 0.001), thinner (28.62 vs. 33.35 kg/m^2^), and have a higher number of areas affected (2.32 vs. 1.38, *p* = 0.02) than those without OCPs. After 12-weeks of treatment, it was observed that the percentage of AN reduction was higher in HS women receiving OCP than in patients without OCP (53.9% vs. 38.42%, *p* = 0.049). It was observed that OCP prescription (β = 3.79, *p* = 0.034) and concomitant therapy (β = 3.91, *p* = 0.037) were independently associated with a higher % AN when controlling for disease duration, concomitant therapy, and treatment with/without OCP (R^2^ = 0.67). The factors potentially associated with the percentage AN reduction at week 12 in HS women treated with OCPs were disease duration (β = −1.327, *p* = 0.052), concomitant therapy (β = 11.04, *p* = 0.079), and HS worsening with the menstrual cycle (β = 10.55, *p* = 0.087). In conclusion, OCPs might be effective for improving AN count in women with HS. Women whose HS worsens in relation to the menstrual cycle and have a shorter disease may benefit more from the therapeutic effect of OCPs.

## 1. Introduction

Hidradenitis suppurativa (HS) is a chronic, recurrent, debilitating inflammatory skin disease of the hair follicle that usually presents after puberty with painful, deep-seated inflamed lesions in the apocrine gland-bearing areas of the body, most commonly the axillae, inguinal, and anogenital regions [1,2]. It has an estimated prevalence rate of around 1% [3], and it disproportionally affects women of childbearing age [4].

The etiopathogeneses of HS seems to be multifactorial but this is not completely understood [5]. HS was originally only regarded as a cutaneous disorder but, today, a growing body of evidence links HS with several dermatological [6] and non-dermatological disorders [7,8,9]. Thus, its inclusion in autoinflammatory systemic diseases should be mandatory for both clinicians and scientists [10,11]. It has been observed that sex hormones may play a role in its pathogenesis [12], as disease severity may vary in intensity according to the menstrual cycle and pregnancy [13]. In fact, HS outbreaks are usually perimenstrual, and they may decrease during pregnancy and menopause [14]. Levels of estradiol and progesterone decrease during the premenstrual period, which may indicate that HS is influenced by fluctuations in the hormones involved in the menstrual cycle [15]. This correlation could be explained because estrogen may inhibit proinflammatory Th1 and Th17 cytokines and thus favor an immunosuppressive environment [13].

Moreover, parallels have been drawn between HS and acne vulgaris [16]. Androgens increase the keratinization of the hair follicle, leading to follicular obstruction, favoring the appearance of outbreaks and exacerbations of HS [17]. Weight gain is also known to contribute to HS pathogenesis [18], and it has been observed that high levels of testosterone and low levels of sex hormone-binding globulin (SHGB) are associated with high BMI [19].

While decreasing levels of progesterone and estrogen seem to coincide with disease flares in premenopausal women, this association is speculative and requires experimental confirmation [12]. To date, recommendations on hormonal therapies are based on limited evidence [20] and there is a need to establish the role of antiandrogens as an alternative or concomitant therapy for HS [21]. Patients reporting HS flares around menses or with features of polycystic ovarian syndrome may be more likely to benefit [22,23]. Therefore, it is important to evaluate whether oral contraceptive pills (OCPs) could improve HS and to assess the kind of women who may benefit from these treatments. Thus, the objectives of this study are (1) to describe the clinical profile of patients receiving oral contraceptive therapy for HS, (2) to assess treatment safety, tolerability and effectiveness at week 12.

## 2. Materials and Methods

Design.

A prospective observational study was conducted between October 2019 and March 2020 in the Hospital Universitario Virgen de las Nieves, Granada, Spain.

Participants.

This study included all patients diagnosed with HS who started oral contraceptive pills (OCPs) for first time at the HS Clinic of Hospital Universitario Virgen de las Nieves, Granada, Spain, which began its activity in February 2017. The recruitment and the assessment of participants were carried out by the chief dermatologist of the HS Clinic. Subjects were identified by replacing their original subject ID with a new random subject ID.

Inclusion criteria: Women of childbearing age (15–49 years) with menstrual cycles who were prescribed OCPs for the treatment of HS. Concomitant treatment with clindamycin gel 1% twice daily or oral doxycycline 100 mg twice daily depending on the severity of HS was prescribed according to current guidelines [24]. For each participant, a control was chosen. A control was the first HS patient that attend our HS clinic after a case inclusion and that was only receiving clindamycin or doxycycline, without OCPs treatment.

Exclusion criteria: Women with HS who did not sign the written consent form or young women under 18 whose legal representative did not sign the informed consent. Climacteric women. Women who were already taking OCPs before visiting our HS Unit.

Variables of interest.

Main variables of interest:(1)Type of OCP prescribed. Combined OCP: one that includes a combination of estrogen and progestin. Non-combined OCP: one that only includes progestin. The type of OCP prescribed was selected according to strict clinical criteria, avoiding combined OCP use in patients with thrombotic risk [25,26]. All women in the combined OCP group were receiving ethinyl estradiol 0.02 + drospirenone 3 mg/24 h with 7 days rest per month, and all women in the non-combined OCP only received the progestogen desogestrel 75 mcgr/24 h.(2)Treatment effectiveness at week 12 assessed by:
The main outcome of interest was the percentage of total abscess and inflammatory nodule (AN) reduction [27].The reduction in 55% of International Hidradenitis Suppurativa Severity Score System (IHS4)-55. The IHS4 score was calculated by the number of nodules (multiplied by 1) plus the number of abscesses (multiplied by 2) plus the number of draining tunnels (multiplied by 4). Recently, in the 2021 Europea Hidradenitis Suppurativa Forum the 55% reduction of IHS4 has been proposed as a binary outcome with good correlation with HiSCR, which do not require a minimum baseline AN [28].Subjective severity improvement. This was evaluated by a numerical rating system (NRS), for pain, odor, suppuration, itching and general condition. It consists of a line numbered from 0 to 10, where 0 is absence of pain or discomfort, and 10 is the maximum degree of pain or discomfort [29].

Other variables of interest:

Clinical, sociodemographic and biometric variables were recorded by means of clinical interview and physical examination. Sociodemographic characteristics included sex, age, civil status, level of education, family history of HS, body mass index (BMI), smoking habit, alcohol consumption and comorbidities. Clinical features included age at HS onset, disease duration, Hurley stage, number of affected areas, nodules, abscesses and draining tunnels count and concomitant treatments.

Ethics.

All patients agreed with the treatment regimen and signed a written consent form to use their personal data for the present study. This study was approved by the Ethics Committee of the Hospital Universitario Virgen de las Nieves and is in accordance with the Helsinki Declaration.

Statistical analysis.

Descriptive statistics were used to evaluate the characteristics of the sample. The Shapiro-Wilk test was used to check the normality of the variables. Continuous data were expressed as mean ± standard deviation (SD) or as the median (25th–75th percentile). The absolute and relative frequency distributions were estimated for qualitative variables. The student’s *t*-test or the Wilcoxon test were used to compare nominal and continuous data, and the χ^2^ test or Fisher’s exact test were applied to nominal data where necessary. Multivariate logistic regression analyses were performed to independently assess the potential effect of OCPs on disease severity. Epidemiological and statistical criteria were used to model variable selection. Significance was set for all tests at two tails, *p* < 0.05. Statistical analyses were performed using JMP version 14.1.0 (SAS institute, Cary, NC, USA).

## 3. Results

### 3.1. Baseline Characteristics

One hundred participants were included in the study: 50 HS women receiving OCPs and 50 without OCPs. Their sociodemographic and clinical features are summarized in Table 1. HS patients with OCPs treatment were younger (31.7 vs. 40.9 years, *p* < 0.001) thinner (28.62 vs. 33.35 kg/m^2^) and have a higher number of areas affected (2.32 vs. 1.38, *p* = 0.02) than those without OCPs.

### 3.2. Effectiveness and Safety of Oral Contreceptives Pills and Factors Potentially Related to the Reduction in the Number of Abscesses and Nodules at 12 Weeks of Treatment

After the follow-up, it was observed that the percentage of AN reduction was higher in HS women receiving OCP than in patients without OCP (53.9% vs. 38.42%, *p* = 0.049). No differences were observed in IHS4-55, Figure 1. Factors associated with a higher % AN reduction are shown in Table 2.

A multivariate regression model was constructed, adjusted for disease duration, concomitant therapy and treatment with/without OCP. It was observed that OCP prescription (β = 3.79, *p* = 0.034) and concomitant therapy (β = 3.91, *p* = 0.037) were independently associated with a higher % AN reduction at week 12 when controlling for the other variables included in the model (R^2^ = 0.67).

The factors potentially associated with the percentage AN reduction at week 12 in HS women treated with OCPs are reflected in Table 3. A multivariate regression model was constructed, adjusted by disease duration, concomitant therapy, number of affected areas and HS worsening with the menstrual cycle. It was observed that disease duration (β = −1.327, *p* = 0.052), concomitant therapy (β = 11.04, *p* = 0.079) and HS worsening with the menstrual cycle (β = 10.55, *p* = 0.087) were almost independently associated with a higher % AN reduction at week 12 when controlling for the other variables included in the model (R^2^ = 0.20).

### 3.3. Type of Oral Contraceptive Pills

Thirty-three women received combined OCPs and 17 non-combined OCP. Demographic and clinical features depending on OCP type are described in Table 4. Women prescribed non-combined OCPs were older than those prescribed combined OCPs (42.58 vs. 26.09, <0.001) and had a longer disease duration (19.09 vs. 9.87 years, *p* = 0.001). Women taking non-combined OCPs were also more frequently active smokers (76.5% vs. 24.26%, *p* < 0.001). Moreover, women taking non-combined OCPs were in Hurley stage III and had severe disease assessed by IHS4 and global VAS.

A multivariate logistic regression model was constructed, adjusted for age, Hurley stage, smoking habit and HS worsening with the menstrual cycle to assess independent factors associated with the type of OCP prescribed. It was observed that for every year that the patient’s age increases, the probability of being prescribed a non-combined OCP instead of a combined OCP was 1.37 times higher. Moreover, for every Hurley stage increase, the probability of being prescribed a non-combined OCP instead of a combined OCP was 11.73 times higher, Table 5.

## 4. Discussion

To the best of our knowledge, this study describes for the first time a higher AN reduction in HS women treated with OCPs and describe the clinical profile of these patients. We found two patient profiles: those receiving combined OCPs and those receiving progestin only. We observed an improvement in inflammatory activity after using OCPs and identified clinical factors potentially associated with a greater treatment effect.

The profile of HS patients being treated with OCPs is that of a young overweight woman, with a family history of HS and mild-moderate disease. The mean age of the population studied is between 20 and 30 years, similar to other HS populations [30], as OCPs should be avoided in older or post-menopausal women [31]. Smoking habit, a risk factor for developing HS, but also a relative contraindication for OCP use, was relatively frequent in our population (44%), in agreement with previous reports [32]. Nevertheless, the thrombotic risk is determined by estrogen, so older women who smoke were mainly prescribed non-combined OCPs. Regarding educational level, the rate of highly educated people in our study was low. This may be related to the disease chronicity in which relapses are frequent, which may have a great impact on the patient’s quality of life and influence their social, occupational, and psychological lives [33]. Over half of our population was overweight. Obesity is also a risk factor for developing HS [34] because adipocytes stimulate the overproduction of proinflammatory cytokines and HS relapses increase because of the mechanical irritation, occlusion and maceration. Two-fifths of our population had a family history of HS. Reports have previously shown a probable genetic component in HS, as having a family history of the disease doubles the risk of suffering from HS [33]. Moreover, we found that more than 90% of the patients receiving OCPs were in stage I and II of the Hurley classification. This is explained because only concomitant treatment with topical clindamycin or oral doxycycline was allowed, therapies used in mild-moderate disease [20].

Our study shows a decrease in AN count in patients undergoing treatment with OCPs and oral antibiotics for 12 weeks without serious adverse effects compared to those women not receiving OCPs. We only observed a reduction in AN but not in IHS4-55. IHS4-55 is a scoring tool that considers both AN and draining tunnel count [28]. The effect of OCPs might be only on inflammatory lesions (nodules and abscesses) and do not work on draining tunnels, explaining the differences in AN but not in IHS4-55. Drospirenone has anti-mineralocorticoid and anti-androgenic power that may reduce the symptoms of HS [21]. Although some studies show that OCPs in combination with drospirenone is more effective than when prescribed alone [35], similar reduction in AN count between the two OCPs were found in our study. This may be because contraceptives combined with drospirenone show efficacy over a longer time period, from 6 months to 1 year [35]. Moreover, one of the benefits of OCPs, whether anti or proandrogenic, is to avoid hormonal imbalances and the worsening associated with menstruation, which may explain similar results using both OCPs.

Factors independently associated with an increase in OCP effectiveness were a shorter HS duration, worsening in relation to the menstrual cycle and being receiving a concomitant therapy. A long disease duration is related to the presence of scars and draining tunnels [36], when OCPs are less effective. It was shown that the worsening of HS with the menstrual cycle was almost independently related to the percentage of AN reduction in patients with OCP treatment. OCPs inhibit the hypothalamic-pituitary axis, repressing basal plasma levels of follicle-stimulating hormone (FSH) and luteinizing hormone (LH) and eliminating oscillation cyclical surges in FSH, LH and estrogen levels [37]. Intraindividual variations in the menstrual cycle [38] may explain why some women with HS get worse in relation to their menstrual cycle and improve more with OCP use, as those women that worsen in relation to the menstrual cycle could have higher level levels of progesterone and estrogen before menstruation [12]. It is possible that the assessment of a hormonal profile could help to select the women who would best respond to OCPs. Moreover, the genetic heterogeneity in HS [39] and the differential response to therapies influenced by patients’ comorbidities [40] may justify variations in womens’ OCP response.

Clinicians should also consider that women with painful menstrual periods and menstrual migraines may take non-steroidal anti-inflammatory drugs (NSAIDs), a possible trigger factor in HS. So, treating this syndrome with OCPs may remove these symptoms and decrease the menstrual use of NSAIDs and HS exacerbation [41,42].

Regarding OCP type, combined OCPs were more frequently prescribed than progesterone-only. We found that the use of non-combined OCPs was independently associated with higher age and more severe disease. Combined OCPs should be avoided in women with risk factors of thrombosis: >35 years or smokers [31]. So, our data may be explained because older people [43] and higher Hurley stages [44] are associated with greater cardiovascular risk.

The results of this study should be considered in light of some limitations: (1) the small sample size, (2) the short follow-up, (3) concomitant treatment allowed (both for HS as clindamycin and doxycycline and NSAIDs), (4) a higher percentage of Hurley III patients in the control group, although no statistically significant differences were found between groups and the AN count at baseline was similar between patients with OCPS and without OCPs.

## 5. Conclusions

In conclusion, to the best of our knowledge, this is the first report to describe the profile of patients receiving OCPs in a real-life setting and finds that women whose HS worsens in relation to the menstrual cycle and have a shorter HS duration may benefit more from the therapeutic effect of OCPs. The increase in the number of effective therapies and the needs of HS patients makes necessary a multidisciplinary approach, including dermatologist, gynecologists, endocrinologists, and surgeons, to improve the management of HS patients.

## Figures and Tables

**Figure 1 life-11-00697-f001:**
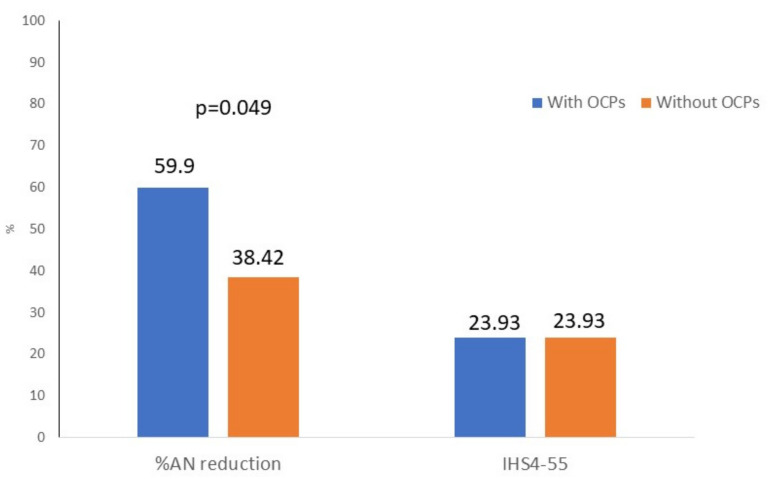
Differences in effectiveness evaluated by percentage of AN count reduction and IHS4-55.

**Table 1 life-11-00697-t001:** Baseline features of the sample.

		Patients with OCPs Treatment (*n* = 50)	Patients without OCPs Treatment (*n* = 50)	*p*
Baseline demographic features				
Age (years)		31.7 (10.41)	40.9 (14.06)	*p* < 0.001
Family history (yes)		40% (20)	34% (17)	0.679
Disease duration (years)		13.04 (8.80)	12.78 (10.04)	0.891
BMI (kg/m^2^)		28.62 (6.41)	33.35 (7.43)	0.001
Smoking habit (yes)		42% (21)	42% (21)	1.00
Baseline clinical features of the sample				
Hurley stage	I	42% (21)	28% (14)	0.275
	II	46% (23)	52% (26)	
	III	12% (6)	20% (10)	
Number of affected areas		2.32 (1.11)	1.78 (1.17)	0.020
IHS4 at baseline		6.58 (4.66)	8.24 (6.81)	0.158
AN at baseline		3.24 (2.33)	3.5 (3.02)	0.631
VAS for pain		4.62 (3.40)	4.88 (3.46)	0.706
VAS for malodor		2.86 (3.39)	3.74 (3.66)	0.215
VAS for itching		4.32 (3.76)	4.44 (3.15)	0.863
VAS for suppuration		3.42 (3.73)	4.46 (3.38)	0.147
VAS global		4.92 (2.66)	5.42 (2.96)	0.376
Concomitant therapy	Oral doxycycline	60% (30)	64% (32)	0.680
	Clindamycin gel	40% (20)	36% (18)	

AN, total abscess and inflammatory nodule count; BMI, Body Mass Index; IHS4, Hidradenitis Suppurativa Severity Score System; OCP, oral contraceptive pills; VAS, Visual Analog Scale. Data are expressed as relative (absolute) frequencies and means (standard deviation (SD)).

**Table 2 life-11-00697-t002:** Factors associated with % AN reduction at week 12.

		% AN Reduction	*p*
OCP treatment	Yes	53.90 (5.49)	0.049
	No	38.42 (5.49)	
Age ^b^		−0.48 (0.30)	0.111
Family history ^a^	Yes	42.79 (6.50)	0.515
	No	48.14 (4.98)	
Disease duration ^b^		−0.64 (0.41)	0.129
BMI ^b^		−0.42 (0.54)	0.439
Smoking habit ^a^	Yes	46.15 (6.11)	0.999
	No	46.17 (6.11)	
Concomitant therapy ^a^	Oral doxycycline	51.83 (4.94)	0.066
	Clindamycin gel	36.91 (6.32)	
Hurley stage ^a^	I	54.76 (6.34)	0.2647
	II	40.60 (5.61)	
	III	44.38 (9.82)	
Number of affected areas ^b^		3.47 (3.39)	0.310

AN, total abscess and inflammatory nodule count; AN12, AN count reduction at week 12; BMI, Body Mass Index; OCP, oral contraceptive pills. ^a^ The Student’s t test for independent samples or the ANOVA test was used to evaluate the association between AN count reduction week 12 (AN12) and categoric variables, data are expressed as a mean (standard deviation). ^b^ Simple linear regression was performed to evaluate the association between AN12 and continuous variables, the data are expressed as a beta coefficient (standard deviation).

**Table 3 life-11-00697-t003:** Factors potentially associated with % AN reduction in women treated with OCPs.

		AN12	*p*
Age ^b^		−0.30 (0.57)	0.593
Family history ^a^	Yes	53.92 (9.25)	0.998
	No	53.89 (7.55)	
Disease duration ^b^		−1.21 (0.65)	0.067
Irregular menses ^a^	Yes	56.53 (6.76)	0.445
	No	46.41 (11.40)	
Worsening in relation to the menstrual cycle ^a^	Yes	69.22 (6.72)	0.083
	No	47.33 (0.083)	
BMI ^b^		−0.72 (0.92)	0.433
Smoking habit ^a^	Yes	49.76 (8.99)	0.548
	No	56.90 (7.65)	
OCP type ^a^	Combined OCP	59.60 (7.06)	0.173
	Non-combined OCP	42.84 (9.83)	
Concomitant therapy ^a^	Oral doxycycline	57.78 (7.50)	0.417
	Clindamycin gel	48.08 (9.18)	
Hurley stage ^a^	I	59.52 (9.03)	0.620
	II	51.96 (8.63)	
	III	41.67 (16.89)	
Number of affected areas ^b^		−5.46 (5.24)	0.303

AN, total abscess and inflammatory nodule count; AN12, AN count reduction at week 12; BMI, Body Mass Index; OCP, oral contraceptive pills. ^a^ The Student’s t test for independent samples or the ANOVA test was used to evaluate the association between AN count reduction week 12 (AN12) and categoric variables, data are expressed as a mean (standard deviation). ^b^ Simple linear regression was performed to evaluate the association between AN12 and continuous variables, the data are expressed as a beta coefficient (standard deviation).

**Table 4 life-11-00697-t004:** Baseline features of the sample depending on the type of oral contraceptive pills prescribed.

		Combined OCP (*n* = 33)	Non-Combined OCP (*n* = 17)	
Baseline demographic features				
Age (years)		26.09 (7.26)	42.59 (5.95)	<0.001 *
Educational level	Mandatory incomplete	12.1% (4/33)	17.6% (3/17)	0.755
	Mandatory	57.6% (19/33)	47.1% (8/17)	
	Superior	30.3% (10.33)	35.3% (6/17)	
Family history (yes)		36.4% (12/33)	47.1% (8/17)	0.465
Disease duration (years)		9.48 (5.23)	19.94 (10.27)	0.001 *
Irregular menses (yes)		27.3% (9/33)	23.5% (4/17)	0.775
Worsening in relation to the menstrual cycle (yes)		27.3% (9/33)	29.4% (5/17)	0.873
BMI (kg/cm^2^)		27.95 (5.79)	29.92 (7.50)	0.308
Smoking habit (yes)		24.26% (8/33)	76.5% (13/17)	<0.001 *
Baseline clinical features of the sample				
Hurley stage	I	57.6% (19/33)	11.8% (2/17)	0.001 *
	II	39.4% (13/33)	58.8% (10/17)	
	III	3% (1/33)	29.4% (5/17)	
Number of affected areas		2.18 (1.13)	2.59 (1.06)	0.323
IHS4 at baseline		5.33 (4.34)	9.00 (4.42)	0.007 *
AN at baseline		3.06 (2.90)	2.71 (1.40)	0.56
Number of inflammatory nodules		2.00 (2.19)	1.12 (1.22)	0.074
Number of abscesses		1.06 (1.39)	1.59 (1.23)	0.193
Number of draining tunnels		0.30 (0.59)	1.18 (1.13)	0.007 *
VAS for pain		3.48 (3.12)	6.82 (2.83)	0.001 *
VAS for malodor		4.76 (3.63)	1.88 (2.83)	0.003 *
VAS for itching		3.39 (3.57)	6.12 (3.53)	0.014 *
VAS for suppuration		5.71 (3.70)	2.24 (3.19)	0.001 *
VAS global		4.12 (2.63)	6.47 (1.97)	0.001 *
Concomitant therapy	Oral doxycycline	57.6% (19/33)	64.7% (11/17)	0.626
	Clindamycin gel	42.4% (14/33)	35.3% (6/17)	

AN, total abscess and inflammatory nodule count; BMI, Body Mass Index; IHS4, Hidradenitis Suppurativa Severity Score System; OCP, oral contraceptive pills; VAS, Visual Analog Scale. Data are expressed as relative (absolute) frequencies and means (standard deviation (SD). The Student’s t test for independent samples was used to compare continuous variables and the chi-square test or Fisher’s exact test, as appropriate, were applied to compare categoric data. Two-tailed * *p* < 0.05 was considered statistically significant in all tests.

**Table 5 life-11-00697-t005:** Multivariate analysis of the factors related to OCPs type prescribed.

Variable	aOR	*p*
Age (years)	1.37	0.007 *
Hurley stage	11.73	0.040 *
Smoking habit	3.41	0.328
Worsening in relation to the menstrual cycle	0.63	0.743
R^2^ Cox y Snell = 0.59		

A logistic regression model was constructed to determine the variables influencing of the type of OCP prescribed (dependent variable) adjusted for age (continuous), Hurley stage (I, II, III), smoking habit (no, yes) and HS worsening in relation to the menstrual cycle (no, yes). Adjusted odds ratios (aOR) are presented. Two-tailed * *p* < 0.05 was considered statistically significant in all tests.

## Data Availability

The data presented in this study are available on request from the corresponding author.

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
