# Peer review of "The Role of Oral Contraceptive Pills in Hidradenitis Suppurativa: A Cohort Study"

_life, 2021, doi:10.3390/life11070697_

Round 1
Reviewer 1 Report
I read the replies and comments and I am satisfied with the revisions.Author Response
Thank you very much for your comments
Reviewer 2 Report
The presentation of the results have been much improved.
However, there is still an obvious bias : the control group includes more Hurley 3 patients, who are much less likely to improve their lesions, since they are more severe, helping the comparison to be in favour of the OCP group. Therefore, the authors should at least state this in the discussion.
On top of this, it sounds very surprising that this predominantly Hurley 3 control group would receive only topical clindamycin or doxycyclin, which is NOT the currently recommended treatment for Hurley 3 patients. The authors should explain why this group did not receive a more potent treatment.
Author Response
The presentation of the results have been much improved.
Thank you very much.
However, there is still an obvious bias : the control group includes more Hurley 3 patients, who are much less likely to improve their lesions, since they are more severe, helping the comparison to be in favor of the OCP group. Therefore, the authors should at least state this in the discussion.
Thank you for your comment. Following your recommendations, we have included the following sentence in the study limitations: a higher percentage of Hurley III patients in the control group, although no statistically differences were found between groups and the AN count at baseline was simi-lar between patients with OCPS and without OCPs. However, the risk of bias is very low for different reasons. On the one hand, as you known Hurley staging refers only to structural damage with independence of inflammatory load, it is surgical staging system. A patient can have a Hurley I stage and have a more severe disease than a patient with Hurley III in terms of inflammation. Inflammatory severity can be assessed in different ways for example with the AN count. AN count at baseline were similar between groups in our study (3.24 vs 3.5, p=0.631). The aim of the medical treatment, like OCP is to diminish inflammatory load. On the other hand, there are 12% (6/50) HS patients with Hurley III in the OCPs group and 20% (10/50) Hurley III patients in the group without OCPs with no statistically differences between groups (p=0.275).
On top of this, it sounds very surprising that this predominantly Hurley 3 control group would receive only topical clindamycin or doxycyclin, which is NOT the currently recommended treatment for Hurley 3 patients. The authors should explain why this group did not receive a more potent treatment
Thank you for your comment. We do not understand why Hurley III is predominant in the control group with only 20% of the patients (10/50). This group is in fact the less common being the majority of the patients Hurley I and II 80% (40/50). We believe that reporting Hurley Stages is always recommended as they give information about structural damage as we have previously stated. For example, I have a patient with Hurley I disease, AN count 16 and other patient with Hurley III with mostly rope-like scars and AN count of 4. Treating the patient with Hurley I with topical clindamycin of doxycycline will be not the advisable, it will be undertreatment. This type of patient after failure or intolerance to a course of antibiotics, will be maybe candidate for more potent treatments like biologic. The patient with Hurley III has a low inflammatory load topical clindamycin or doxycycline will be a good treatment for the patient combined with surgery of the different involved areas. As you can see in our sample, medical treatment is chosen based on inflammatory load, AN count at baseline were similar between groups in our study (3.24 vs 3.5, p=0.631). Our patients with Hurley III will also receive surgery in the future. Combining surgical and medical treatment is critical for Hurley II and III patients since the target for different elements of the disease. We strongly believe that our rationale for treatment decisions in our HS patients is ethic, adjusted to inflammatory and structural severity of each patient and in line with the most advanced scientific knowledge.
This manuscript is a resubmission of an earlier submission. The following is a list of the peer review reports and author responses from that submission.
Round 1
Reviewer 1 Report
In this study the authors present a clinical profile for 61 HS patients treated with OCP, when 50 of them completed the treatment after 3 months. I have some remarks:
why did the authors use the AN count and not a validated system such as IHS4 or DLQI for their primary endpoint?
Was the OCP started for the indication HS? Was it administered/prescribed froma gynecologist or a dermatologist? The antiandrogen effect of drosperinon is well known for acne, desorgestrel has however a proandrogen action? The AN reduction was absolutely the same. How do the authors see that? Please comment (slight modification of the discussion in the paragraph about the HPA axis)
How do you explain the lack of significance between patients who finished treatment and the ones, which interrupted the therapy before week 12?
Overall I believe that the clear clinical profile of these HS women belongs in the literature and the fact that even PROandrogenic OCPs kann help with HS and might have less thrombosis index. My personal speculation is that the extension of the study with the primary endpoint on week 24 would provide a a more clear significance for more than one of the studied variables.
Author Response
In this study the authors present a clinical profile for 61 HS patients treated with OCP, when 50 of them completed the treatment after 3 months. I have some remarks:
Thank you for the comments.
why did the authors use the AN count and not a validated system such as IHS4 or DLQI for their primary endpoint?
We used the AN count because of nodule and abscess are the first lesions to respond to treatment. In fact, draining fistulas do not respond to adalimumab until 16 weeks treatment, so it would be difficult to find differences between groups in IHS4, as it includes draining fistulas count.
Was the OCP started for the indication HS? Was it administered/prescribed froma gynecologist or a dermatologist? The antiandrogen effect of drosperinon is well known for acne, desorgestrel has however a proandrogen action? The AN reduction was absolutely the same. How do the authors see that? Please comment (slight modification of the discussion in the paragraph about the HPA axis)
OCPs were started for treating HS. Women who were already taking OCPs before visiting our HS Unit were excluded as mentioned in the exclusion criteria section. OCPs were prescribed by a dermatologist. Non-combined OCPs, desogestrel, was use if the patient had thrombotic risk. Similar reduction in AN count using each OCPs may be explained because one of the benefits of OCPs, whether anti or proandrogenic, is to avoid hormonal imbalances and the worsening associated with menstruation, and that therefore, at least in short term, there may be no difference. We have changed the sentence in the discussion and have added the possible reason for it: Although some studies show that OCPs in combination with drospirenone is more effective than when prescribed alone [35], similar reduction in AN count between the two OCPs were found in our study. This may be because contraceptives combined with drospirenone show efficacy over a longer time period, from 6 months to 1 year [35]. Moreover, one of the benefits of OCPs, whether anti or proandrogenic, is to avoid hormonal imbalances and the worsening associated with menstruation, what may explain similar results using both OCP.
How do you explain the lack of significance between patients who finished treatment and the ones, which interrupted the therapy before week 12?
We have only provided differences in clinical and sociodemographic characteristics between both group at baseline because we wanted to look for reasons that may lead to treatment non-compliance. Unfortunately, we did not find any clinical predictor, maybe due to reduced sample size. We did not analyse efficacy in patients that did non-comply treatment and this data is not included in the manuscript.
Overall I believe that the clear clinical profile of these HS women belongs in the literature and the fact that even PROandrogenic OCPs kann help with HS and might have less thrombosis index. My personal speculation is that the extension of the study with the primary endpoint on week 24 would provide a a more clear significance for more than one of the studied variables.
We also think that a long follow-up would have provided more clear significance. We would have liked to follow-up longer our participants. Nevertheless, the COVID-19 outbreak makes it difficult to make a close patient monitoring. As we observed differences in 12-weeks treatment, it is expected more differences were found after 24-weeks follow-up. Our study highlights for the first time the role of OCP in HS. As mentioned in limitation, longer follow-up and studies with controls groups are needed to support our results. We are now working in a multi-centre study regarding different OCPs use with also a control group.
Reviewer 2 Report
This is the first work in the current times, which systematically analysed the effectiveness of OCP in women with HS and the first work, which studied the effectiveness of OCP without combination with enhanced antiandrogen dosis. The work could confirm several aspects of OCP treatment in HS, which were up-to-now suggested due to individual experience, providing in addition the required level of evidence for including OCP in guidelines for HS treatment.
Line 140: “…of the women with HS had a mild or moderate disease as assessed by Hurley stage.”: The authors should modify the statement in “…of the women with HS were in Hurley stage I to II.”, since classification of HS in mild to moderate disease is not compatible with Hurley’s classification.
lines 166-167: “…had severe disease assessed by Hurley stage, IHS4 and global VAS.” should be modified in “…were in Hurley stage III and had severe disease assessed by IHS4 and global VAS.” for the same reasons.
Author Response
Reviewer 2
This is the first work in the current times, which systematically analysed the effectiveness of OCP in women with HS and the first work, which studied the effectiveness of OCP without combination with enhanced antiandrogen dosis. The work could confirm several aspects of OCP treatment in HS, which were up-to-now suggested due to individual experience, providing in addition the required level of evidence for including OCP in guidelines for HS treatment.
Thank you for the comments.
Line 140: “…of the women with HS had a mild or moderate disease as assessed by Hurley stage.”: The authors should modify the statement in “…of the women with HS were in Hurley stage I to II.”, since classification of HS in mild to moderate disease is not compatible with Hurley’s classification.
This sentence has been changed as recommended.
lines 166-167: “…had severe disease assessed by Hurley stage, IHS4 and global VAS.” should be modified in “…were in Hurley stage III and had severe disease assessed by IHS4 and global VAS.” for the same reasons.
This sentence has been changed as recommended.
Reviewer 3 Report
I read with great interest the manuscript titled "Patients with Hidradenitis Suppurativa with perimenstrual worsening can benefit from Oral Contraceptive Pills: A Cohort Study"
It is of great importance in daily clinical practice since menstrual link with HS severity modification remain a frequently asked question.
I have only some minor revisions:
INTRODUCTION
Please Change the first reference with these two more up to date [10.1111/jdv.16677, 10.1159/000507323]
Please add this paragraph: Despite HS was originally regarded as cutaneous disorders a growing body of evidence links HS with several dermatological [10.1007/s00403-020-02105-x] and non-dermatological disorders [10.1001/jamadermatol.2020.5087, 10.4254/wjh.v11.i4.391, 10.1007/s11739-017-1658-0], thus its inclusion in autoimmune systemic diseases should be mandatory for both clinicians and scientists [10.1136/annrheumdis-2016-210901,PMID: 29742056]
Material and Methods
Please describe more the study design clarifying the study interval, hospitals included, colleagues that evaluated..
Please explain how did you identify the subjects (ICD?)
DISCUSSION
Please introduce the concept of genetic heterogeneity in HS [10.1016/j.jid.2019.10.025] and differential response to the therapy influenced by comorbisties [10.1159/000503606] that may justify your results in the patients' subset included.
Please state clearly the limitations.
Please add future perspectives advocating a multi specialists evaluation (i.e. gynecologists-dermatoogists) of HS patients.
Author Response
Reviewer 3
I read with great interest the manuscript titled "Patients with Hidradenitis Suppurativa with perimenstrual worsening can benefit from Oral Contraceptive Pills: A Cohort Study"
It is of great importance in daily clinical practice since menstrual link with HS severity modification remain a frequently asked question.
I have only some minor revisions:
Thank you for the comments.
INTRODUCTION
Please Change the first reference with these two more up to date [10.1111/jdv.16677, 10.1159/000507323]
The first reference has been changed as recommended.
Please add this paragraph: Despite HS was originally regarded as cutaneous disorders a growing body of evidence links HS with several dermatological [10.1007/s00403-020-02105-x] and non-dermatological disorders [10.1001/jamadermatol.2020.5087, 10.4254/wjh.v11.i4.391, 10.1007/s11739-017-1658-0], thus its inclusion in autoimmune systemic diseases should be mandatory for both clinicians and scientists [10.1136/annrheumdis-2016-210901,PMID: 29742056]
This paragraph has been added in the introduction.
Material and Methods
Please describe more the study design clarifying the study interval, hospitals included, colleagues that evaluated..
We have added more information regarding the study design. A prospective observational study was conducted between October 2019 and March 2020 in the Hospital Universitario Virgen de las Nieves, Granada, Spain. The recruitment and the assessment of participants was carried out by the chief dermatologist of the HS Clinic.
Please explain how did you identify the subjects (ICD?)
Subjects were identified by replace their original subject ID with a new random subject ID. This information has been added in the text.
DISCUSSION
Please introduce the concept of genetic heterogeneity in HS [10.1016/j.jid.2019.10.025] and differential response to the therapy influenced by comorbisties [10.1159/000503606] that may justify your results in the patients' subset included.
We have included that the genetic heterogeneity and the comorbidities may influence patients’ response. The following sentence has been added: Moreover, the genetic heterogeneity in HS [38] and the differential response to therapies influenced by patients’ comorbidities [39] may justify variations in women’ OCP response.
Please state clearly the limitations.
The limitations have been rephrased to clarify them: 1) the small sample size, 2) the short follow-up, 3) the absence of a control group, 4) concomitant treatment allowed (both for HS as clindamycin and doxycycline and NSAIDs).
Please add future perspectives advocating a multi specialists evaluation (i.e. gynecologists-dermatoogists) of HS patients.
We have added in the conclusion that The increase in the number of effective therapies and the needs of HS patients makes necessary a multidisciplinary approach, including dermatologist, gynecologists, endocrinologist and surgeons, to improve the management of HS patients.
Reviewer 4 Report
This study attempts to look at the interest of OCP in women of child-bearing age affected with HS.
The study design is very poor with no control group without OCP.
Two different other treatments, namely antibiotics, are concomitantly prescribed. So the design of this study is for comparing 2 groups having different OCP according to age. “Strictly clinical criteria” for prescribing one regimen instead of the other regimen of OCP are not described. Neither are precisely described which patients received topical clindamycin or doxycyclin.
Since other treatments are concomitantly prescribed, without any control group, no conclusion can be drawn about efficacy of OCP alone. Moreover, a previous RCT by Jemec compared the efficacy of topical clindamycin versus a cyclin in HS and already reported a 30% improvement with both treatments, without any OCP (should be cited).
No attempt is made to precise which patients have polycystic ovaries syndrome, which would be an important information. Indeed, the authors report irregular periods (not precisely defined in the article), but no painful periods. Women with PCOS usually have painful periods and sometimes menstrual migraines, both reasons for which they take NSAIDs. So treating this syndrome with OCP may remove these symptoms, therefore the menstrual use of NSAIDs. NSAIDs may be a triggering factor in HS, as underlined by some authors (Communications at SHSA 2019 and EHSF 2020). This way, stopping NSAIDs by using OCP can be another confounding factor.
What was the interest of comparing the group who completed treatment and the group who did not? Was treatment interrupted because of lack of efficacy?
Recent ref that should probably be cited :
Characterizing perimenstrual flares of hidradenitis suppurativa.
Collier EK, Price KN, Grogan TR, Naik HB, Shi VY, Hsiao JL.Int J Womens Dermatol. 2020 Sep 14;6(5):372-376. doi: 10.1016/j.ijwd.2020.09.002. eCollection 2020 Dec.
Menses, pregnancy, delivery, and menopause in hidradenitis suppurativa: A patient survey.
Fernandez JM, Hendricks AJ, Thompson AM, Mata EM, Collier EK, Grogan TR, Shi VY, Hsiao JL.Int J Womens Dermatol. 2020 Jul 10;6(5):368-371. doi: 10.1016/j.ijwd.2020.07.002. eCollection 2020 Dec.
Author Response
This study attempts to look at the interest of OCP in women of child-bearing age affected with HS.
The study design is very poor with no control group without OCP.
Thank you for the comment. Although we did not include an strictly control group, we included the group of women that were prescribed OCPs and did not take them finding no differences in baseline characteristics between both group.
Two different other treatments, namely antibiotics, are concomitantly prescribed. So the design of this study is for comparing 2 groups having different OCP according to age. “Strictly clinical criteria” for prescribing one regimen instead of the other regimen of OCP are not described. Neither are precisely described which patients received topical clindamycin or doxycyclin.
OCPs were prescribed following clinical criteria as recommended in guidelines. Non-combined OCPs were mainly prescribed for those women having thrombotic risk. Clindamycin and doxycycline were also prescribed following the current guidelines for treating HS. This information has been added in the manuscript: Concomitant treatment with clindamycin gel 1% twice daily or oral doxycycline 100mg twice daily depending on the severity of HS was prescribed according to current guidelines [24]. The type of OCP prescribed was selected according to strictly clinical criteria, avoiding combined OCP use in patients with thrombotic risk [25, 26].
Since other treatments are concomitantly prescribed, without any control group, no conclusion can be drawn about efficacy of OCP alone. Moreover, a previous RCT by Jemec compared the efficacy of topical clindamycin versus a cyclin in HS and already reported a 30% improvement with both treatments, without any OCP (should be cited).
We have added the following sentence as recommended: while previously it has been shown similar effectiveness between clindamycin with systemic tetracycline [38].
No attempt is made to precise which patients have polycystic ovaries syndrome, which would be an important information. Indeed, the authors report irregular periods (not precisely defined in the article), but no painful periods. Women with PCOS usually have painful periods and sometimes menstrual migraines, both reasons for which they take NSAIDs. So treating this syndrome with OCP may remove these symptoms, therefore the menstrual use of NSAIDs. NSAIDs may be a triggering factor in HS, as underlined by some authors (Communications at SHSA 2019 and EHSF 2020). This way, stopping NSAIDs by using OCP can be another confounding factor.
We only included women that were prescribed OCPs for treating HS. We did not include patients that were receiving OCPs previously to the visit to our HS clinic. So, we assume that women with painful periods or having polycystic ovaries were receiving OCPs previously and they were not included in the study or if they were included, they only represent a minor part of our study population. Anyway, we have included this information in the discussion section and the following sentences have been added: Clinicians should also consider that women with painful menstrual periods and menstrual migraines may take non-steroidal anti-inflammatory drugs (NSAIDs), a possible trigger factor in HS. So, treating this syndrome with OCPs may remove these symptoms and decrease the menstrual use of NSAIDs and HS exacerbation. Moreover, we have included this topic as a possible study limitation.
What was the interest of comparing the group who completed treatment and the group who did not? Was treatment interrupted because of lack of efficacy?
The interest to compare clinical and sociodemographic characteristics between both group at baseline is to evaluate if there are some factors that could lead to treatment non-compliance and to explore differences between treatment non-compliance in combined and non-combined OCPs. Reasons for treatment interruption are provided: The reasons given were mood swings, hot flushes, headaches, dizziness, swelling, discomfort, menstrual irregularities or medication forgetfulness. Treatment with-drawal was similar in both OCP type: 22.7% (5/22) of patients using non-combined OCPs and 15.4% (6/39) taking combined OCPs.
Recent ref that should probably be cited :
Characterizing perimenstrual flares of hidradenitis suppurativa.
Collier EK, Price KN, Grogan TR, Naik HB, Shi VY, Hsiao JL.Int J Womens Dermatol. 2020 Sep 14;6(5):372-376. doi: 10.1016/j.ijwd.2020.09.002. eCollection 2020 Dec.
Menses, pregnancy, delivery, and menopause in hidradenitis suppurativa: A patient survey.
Fernandez JM, Hendricks AJ, Thompson AM, Mata EM, Collier EK, Grogan TR, Shi VY, Hsiao JL.Int J Womens Dermatol. 2020 Jul 10;6(5):368-371. doi: 10.1016/j.ijwd.2020.07.002. eCollection 2020 Dec.
Recent ref that should probably be cited :
Characterizing perimenstrual flares of hidradenitis suppurativa.
All these references have been cited as recommended.
Round 2
Reviewer 4 Report
The conclusions of the study saying that “women whose HS worsens in relation to the menstrual cycle may benefit more from the therapeutic effect of OCPs" can in no way be drawn, since no control group has been compared and since patients in all groups are taking medications (ie antibiotics) that themselves have been shown to improve HS and are currently recommended in all international guidelines.
The authors answered that the group without treatment completion is a control group. However, the authors did not state this in the text and did not mention that treatment was taken for a few days only in this group of patients. Moreover, if this is a control group in the authors' mind, then results for efficacy should be compared between these 2 groups, (with the condition that groups should be big enough to be significant), instead of comparing combined OCP and non combined OCP. Unless we have this comparison, conclusions written by the authors about an efficiency of OCP in HS are wrong and efficacy is probably due to antibiotic treatments, as previously demonstrated in the RCT testing doxycycline versus topical clindamycin in HS published by Jemec and al, showing an equal improvement of about 30% in both groups.
The improvement observed CANNOT be attributed to OCP, while these patients were on doxy or topical clindamycin. To conclude that OCP brings a supplementary benefit, the authors should have had at least a control group with only OCP vs OCP + antibiotics.
Smaller details:
P2: HS is not currently considered as an auto-immune disorder, but rather as an auto-inflammatory as stated in both titles and articles of ref 10 and 11, cited by the authors.
P2: “role of OCP in HS severity” is not an appropriate wording, a better formulation would be “whether OCP could improve HS”, HS severity being probably partly genetically determined.
HS is reported to often disappear at menopause, when levels of estrogens and progesterone are low, so the reasons given by the authors for a potential participation of sexual hormones in peri-menstrual flares does not work with disappearance at menopause and neither with male HS.